METHODS AND RESOURCES

# The hair cell analysis toolbox is a precise and fully automated pipeline for whole cochlea hair cell quantification

**Christopher J. Buswinka**[1,2], **Richard T. Osgood**[1], **Rubina G. Simikyan**[1], **David B. Rosenberg**[1], **Artur A. Indzhykulian**[1,2]*

1 Mass Eye and Ear, Harvard Medical School, Boston, Massachusetts, United States of America, 2 Speech and Hearing Bioscience and Technology Program, Harvard University, Cambridge, Massachusetts, United States of America

* inartur@hms.harvard.edu

**Data Availability Statement:** The authors confirm that all data underlying the findings are fully available without restriction. All relevant data are within the paper and its Supporting Information

## Abstract

Our sense of hearing is mediated by sensory hair cells, precisely arranged and highly specialized cells subdivided into outer hair cells (OHCs) and inner hair cells (IHCs). Light microscopy tools allow for imaging of auditory hair cells along the full length of the cochlea, often yielding more data than feasible to manually analyze. Currently, there are no widely applicable tools for fast, unsupervised, unbiased, and comprehensive image analysis of auditory hair cells that work well either with imaging datasets containing an entire cochlea or smaller sampled regions. Here, we present a highly accurate machine learning-based hair cell analysis toolbox (HCAT) for the comprehensive analysis of whole cochleae (or smaller regions of interest) across light microscopy imaging modalities and species. The HCAT is a software that automates common image analysis tasks such as counting hair cells, classifying them by subtype (IHCs versus OHCs), determining their best frequency based on their location along the cochlea, and generating cochleograms. These automated tools remove a considerable barrier in cochlear image analysis, allowing for faster, unbiased, and more comprehensive data analysis practices. Furthermore, HCAT can serve as a template for deep learning-based detection tasks in other types of biological tissue: With some training data, HCAT's core codebase can be trained to develop a custom deep learning detection model for any object on an image.

## Introduction

The cochlea is the organ in the inner ear responsible for the detection of sound. It is tonotopically organized in an ascending spiral, with mechanosensitive sensory cells responding to high-frequency sounds at its base and low-frequency sounds at the apex. These mechanically sensitive cells of the cochlea, known as hair cells, are classified into two functional subtypes: outer hair cells (OHCs) that amplify sound vibrations and inner hair cells (IHCs) that convert these vibrations into neural signals [1]. Each hair cell carries a bundle of actin-rich microvillus-like protrusions called stereocilia. Hair cells are regularly organized into one row of IHCs

files. All code has been hosted on github and is available for download at https://github.com/indzhykulianlab/hcat along with accompanying documentation at https://hcat.readthedocs.io/ The EPL cochlea frequency ImageJ plugin is available for download at: https://www.masseyeandear.org/research/otolaryngology/eaton-peabody-laboratories/histology-core.

**Funding:** This work was supported by NIH R01DC020190 (NIDCD), R01DC017166 (NIDCD) and R01DC017166-04S1 "Administrative Supplement to Support Collaborations to Improve the AI/ML-Readiness of NIH-Supported Data" (Office of the Director, NIH) to A.A.I. and the Speech and Speech and Hearing Bioscience and Technology Program Training grant T32 DC000038 (NIDCD). The funders had no role in study design, data collection and analysis, decision to publish, or preparation of the manuscript.

**Competing interests:** The authors have declared that no competing interests exist.

**Abbreviations:** FP, false positive; GUI, graphical user interface; HCAT, hair cell analysis toolbox; IHC, inner hair cell; NMS, non-maximum suppression; OHC, outer hair cell; TN, true negative; TP, true positive.

and three (rarely four) rows of OHCs within a sensory organ known as the organ of Corti [2]. The OHC stereocilia bundles are arranged in a characteristic V-shape and are composed of thinner stereocilia as compared to those of IHCs. Hair cells are essential for hearing, and deafness phenotypes are often characterized by their histopathology using high-magnification microscopy. The cochlea contains thousands of hair cells, organized over a large spatial area along the length of the organ of Corti. During histological analysis, each of these thousands of cells represents a datum that must be parsed from the image by hand ad nauseam. To accommodate for manual analysis, it is common to disregard all but a small subset of cells, sampling large datasets in representative tonotopic locations (often referred to as base, middle, and apex of the cochlea). To our knowledge, there are two existing automated hair cell counting algorithms to date, both of which have been developed for specific use cases, largely limiting their application for the wider hearing research community. One such algorithm by Urata *et al* [3] relies on the homogeneity of structure in the organ of Corti and fails when irregularities, such as four rows of OHCs, are present. It is worth noting however, that their algorithm enables hair cell detection in 3D space, which may be critical for some applications [4]. Another algorithm, developed by Cortada *et al* [5] does not differentiate between IHCs and OHCs. Thus, each were limited in their application, likely impeding their widespread use [3,5]. The slow speed and tedium of manual analysis poses a significant barrier when faced with large datasets, be that analyzing whole cochlea instead of sampling three regions, or those generated through studies involving high-throughput screening [6,7]. Furthermore, manual analyses can be fraught with user error, biases, sample-to-sample inconsistencies, and variability between individuals performing the analysis. These challenges highlight a need for unbiased, automated image analysis on a single-cell level across the entire frequency spectrum of hearing.

Over the past decade, considerable advancements have been made in deep learning approaches for object detection [8]. The predominant approach is Faster R-CNN [9], a deep learning algorithm that quickly recognizes the location and position of objects in an image. While originally designed for use with images collected by conventional means (camera), there has been success in applying the same architecture to biomedical image analysis tasks [10–12]. This algorithm can be adapted and trained to perform such tasks orders of magnitude faster than manual analysis. We have created a machine learning-based analysis software that quickly and automatically detects each hair cell, determines its type (IHC versus OHC), and estimates cell's best frequency based on its location along the cochlear coil. Here, we present a suite of tools for cochlear hair cell image analysis, the hair cell analysis toolbox (HCAT), a consolidated software that enables fully unsupervised hair cell detection and cochleogram generation.

## Results

### Analysis pipeline

HCAT combines a deep learning algorithm, which has been trained to detect and classify cochlear hair cells, with a novel procedure for cell frequency estimation to extract information from cochlear imaging datasets quickly and in a fully automated fashion. An overview of the analysis pipeline is shown in **Fig 1**. The model accepts common image formats (tif, png, and jpeg), in which the order of the fluorescence channels within the images, or their assigned color, does not affect the outcome. Multi-page tif images are automatically converted to a 2D maximum intensity projection. When working with large confocal micrographs, HCAT analyzes small crops of the image and subsequently merges the results to form a contiguous detection dataset. These cropped regions are set to have 10% overlap along all edges, ensuring that each cell is fully represented at least once. Regions that do not contain any fluorescence above a certain threshold may be optionally skipped, increasing speed of large image analysis while

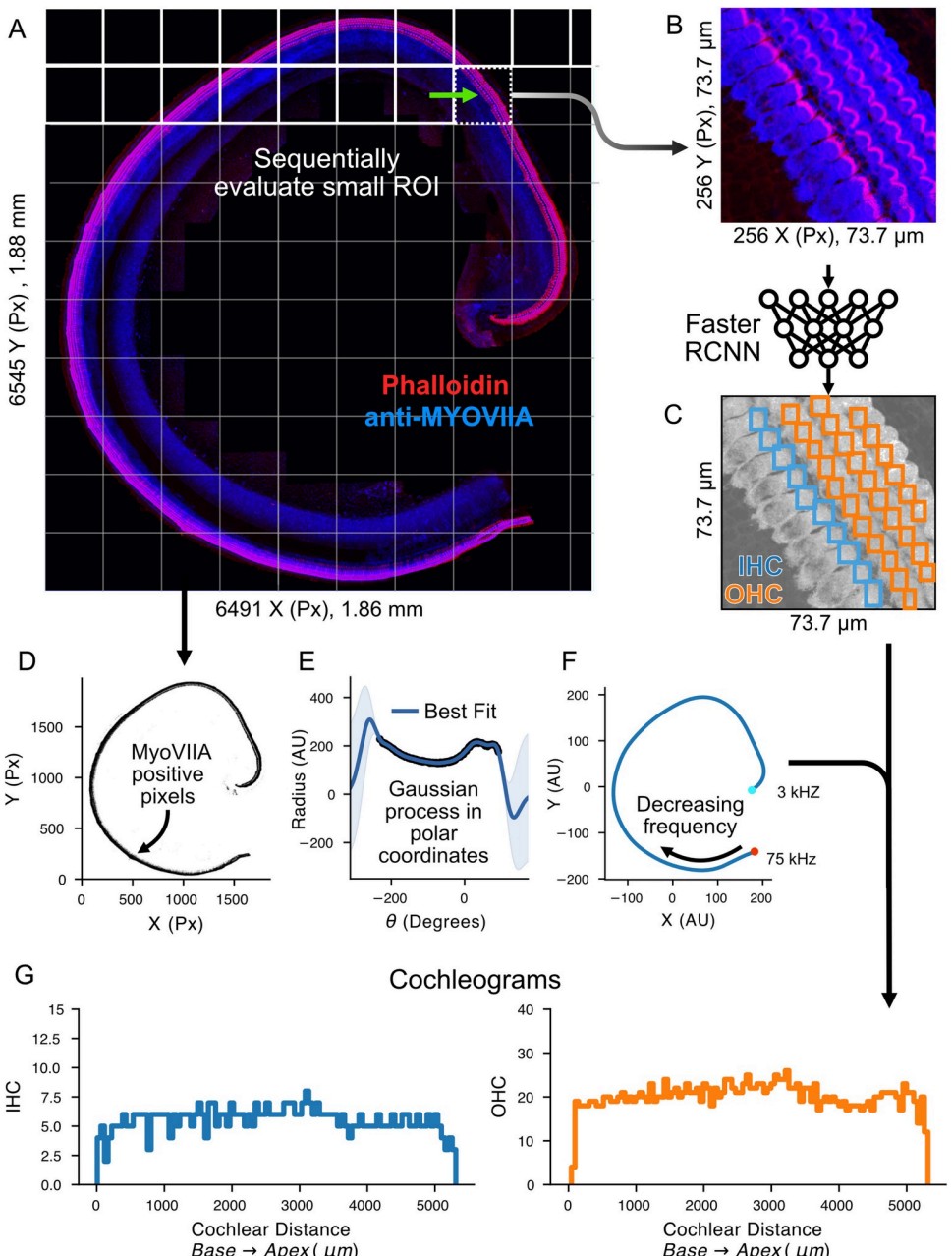

**Fig 1. HCAT analysis pipeline.** An early postnatal wild-type mouse cochlea, dissected in a single contiguous piece, imaged at high magnification (288 nm/px resolution) (**A**) is broken into smaller 256 × 256 px regions and sequentially evaluated by a deep learning detection and classification algorithm (**B**) to predict the probable locations of IHCs and OHCs (**C**). The entire cochlea is then used to infer each cell's best frequency along the cochlear coil. First, all supra-threshold anti-MYO7A-positive pixels are converted to polar coordinates (**D**) and fit by the Gaussian process nonlinear curve fitting algorithm (**E**). The resulting curve is converted back to cartesian coordinates and the resulting line is converted to frequency by the Greenwood function; the apical end of the cochlea (teal circle) is inferred by the region of greatest curl (**F**), and the opposite end of the cochlea is assigned as the basal end (red circle). Cells are then assigned a best frequency based on their position along the predicted curve, and cochleograms (**G**) are generated in a fully automated way for each cell type (IHCs and OHCs), with a bin size by default set to 1% of the total cochlear length. HCAT, hair cell analysis toolbox; IHC, inner hair cell; OHC, outer hair cell.

limiting false positive errors. When the entire cochlea is contained as a contiguous piece (**Fig 1A**), which is common for neonatal cochlear histology, HCAT will estimate the cochlear path and each cell will be assigned a best frequency. Following cell detection and best frequency estimation, HCAT performs two post processing steps to refine the output and improve overall accuracy. First, cells detected multiple times are identified and removed based on a user-defined bounding box overlap threshold, set to 30% by default. The second step, optional and only applicable for whole cochlear coil analysis, removes cells too far from the estimated cochlear path, reducing false positive detections in datasets with suboptimal anti-MYO7A labeling outcomes, such as high background fluorescence levels or instances of nonspecific labeling away from the organ of Corti. As outlined below, for each detection analysis HCAT outputs diagnostic images with overlaid cell-specific data, in addition to an associated CSV data table, enabling further data analysis or downstream post processing, and, when applicable, automatically generates cochleograms.

HCAT is computationally efficient and can execute detection analysis on a whole cochlea on a timescale vastly faster than manual analysis, regularly completing in under 90 s when utilizing GPU acceleration on affordable computational hardware. HCAT is available in two user interfaces: (1) a command line interface that offers full functionality, including cell frequency estimation and batch processing of multiple images or image stacks across multiple folders; and (2) a graphical user interface (GUI), which is user-friendly and is optimized for analysis of individual or multiple images contained within a single folder. The GUI is unable to infer cell's best frequency and is suitable for analysis of small regions of cochlea.

## Detection and classification

To perform cell detection, we leverage the Faster R-CNN [9] deep learning algorithm with a ConvNext [13] backbone trained on a varied dataset of cochlear hair cells from multiple species, at different ages, and from different experimental conditions (**Table 1**, **Fig 2**). Most of the hair cells used to train the detection model were stained with two markers: (1) anti-MYO7A, a hair cell specific cell body marker; and (2) the actin label, phalloidin, to visualize the stereocilia bundle. Bounding boxes for each cell along with class identification labels were manually generated to serve as the ground truth reference by which we trained the detection model (**Fig 2**). Boxes were centered around stereocilia bundles and included the hair cell cuticular plate as

**Table 1. Summary of training data.**

| Laboratory | Number of images | OHC | IHC | Animal | Microscope | Treatment | Age | Labeled Protein | |
|---|---|---|---|---|---|---|---|---|---|
| Artur Indzhykulian, PhD | 45 | 12,959 | 3,706 | Mouse | Confocal | None | P5-P7 | MYO7A | Actin |
| Lisa Cunningham, PhD and Katharine Fernandez, PhD | 77 | 3,424 | 1,290 | Mouse | Confocal | Platinum Compounds | 18–24 wk | MYO7A | Actin |
| Albert Edge, PhD | 2 | 125 | 42 | Mouse | Confocal | None | 8 wk | MYO7A | Actin |
| M. Charles Liberman, PhD | 29 | 894 | 290 | Human | Confocal | None | Adult | MYO7A | ESPN |
| Guy Richardson, PhD and Corne Kros, PhD | 26 | 1,226 | 690 | Mouse | Epifluorescence | Aminoglycosides | P2-P3 | MYO7A | Actin |
| Mark Rutherford, PhD | 5 | 120 | 65 | Mouse | Confocal | None | P30 | MYO7A | Actin |
| Anthony Ricci, PhD | 3 | 120 | 43 | Mouse | Confocal | None | Adult | MYO7A | Actin |
| Basile Tarchini, PhD | 8 | 292 | 97 | Mouse | Confocal | None | P21-P22 | MYO7A | Actin |
| Bradley Walters, PhD | 6 | 904 | 238 | Guinea Pig | Confocal | None | Adult | MYO7A | Actin |
| Total | 201 | 20,064 | 6,461 | | | | | | |

IHC, inner hair cell; OHC, outer hair cell.

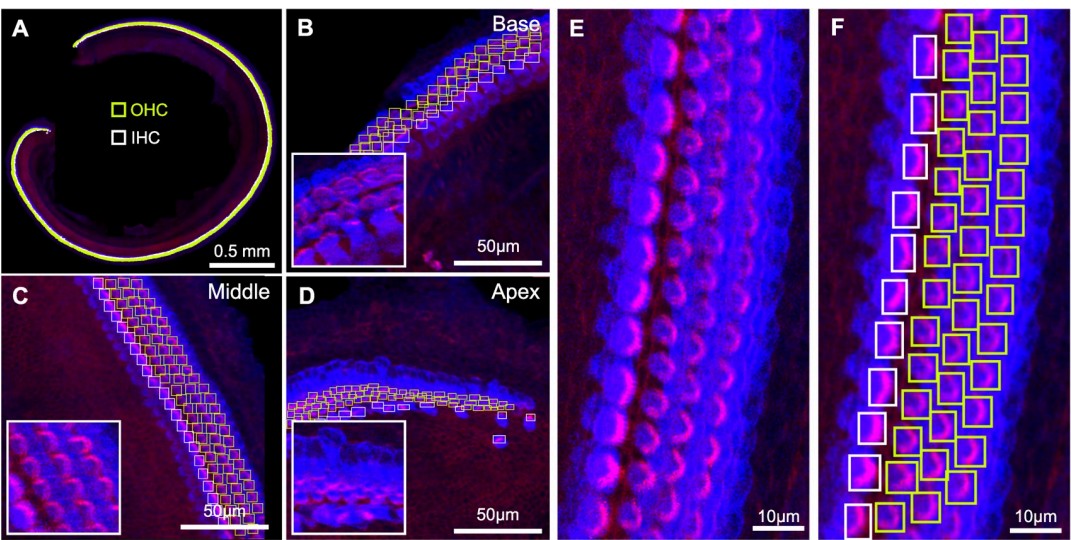

**Fig 2. HCAT detection algorithm training data.** Early postnatal, wild-type murine hair cells in whole cochlea stained against MYO7A (blue) and phalloidin (magenta) were manually annotated by placing either yellow (OHC) or white (IHC) boxes around each stereocilia bundle (**A–D**) and used as training data for the Faster R-CNN deep learning algorithm. All annotated boxes appear as a thin, pale green strip when rendered in (**A**). Hair cells vary in appearance based on tonotopy, with representative regions of the base (**B**), middle (**C**), and apex (**D**) shown here. Since the boundaries between hair cell cytosol (blue) overlap in maximum intensity projection images (**E**), the bounding boxes for each cell were annotated around the stereocilia bundle and cuticular plate of each hair cell (**F**). HCAT, hair cell analysis toolbox; IHC, inner hair cell; OHC, outer hair cell.

these were determined the most robust features per cell in a maximum intensity projection image. The trained Faster R-CNN model predicts three features for each detected cell: a bounding box, a classification label (IHC or OHC), and a confidence score (**Fig 3**).

To limit false positive detections, cells predicted by Faster R-CNN can be rejected based on their confidence score or their overlap with another detection through an algorithm called non-maximum suppression (NMS). To find optimal values for the confidence and overlap thresholds, we performed a grid search by which we assessed model performance at each combination of values and selected values leading to most accurate model performance (**Figs 3E–3G and S1**).

The trained Faster R-CNN detection algorithm performs best on maximum intensity projections of 3D confocal z-stacks of hair cells labeled with a cell body stain (such as anti-MYO7A) and a hair bundle stain (such as phalloidin), imaged at a X-Y resolution of approximately 290 nm/px (**Fig 4D and 4E**). However, the model can perform well with combinations of other markers, including antibody labeling against ESPN, Calbindin, Calcineurin, p-AMPK**α**, as well as following FM1-43 dye loading. HCAT can accurately detect cells in healthy and pathologic cochlear samples, collected within a range of imaging modalities, resolutions, and signal-to-noise ratios. While the pixel resolution requirements for the imaging data are not very demanding, imaging artifacts and low fluorescence signal intensity can limit detection accuracy. Although there is one row of IHCs and three rows of OHCs in most cochlear samples, there are rare instances where two rows of IHCs or four rows OHCs can be seen in normal cochlear samples, the algorithm is robust and largely accurate in such instances (**Fig 4D**).

## Cochlear path determination

For images containing an entire contiguous cochlear coil, HCAT can additionally predict cell's best frequency via automated cochlear path determination. To do this, HCAT fits a Gaussian

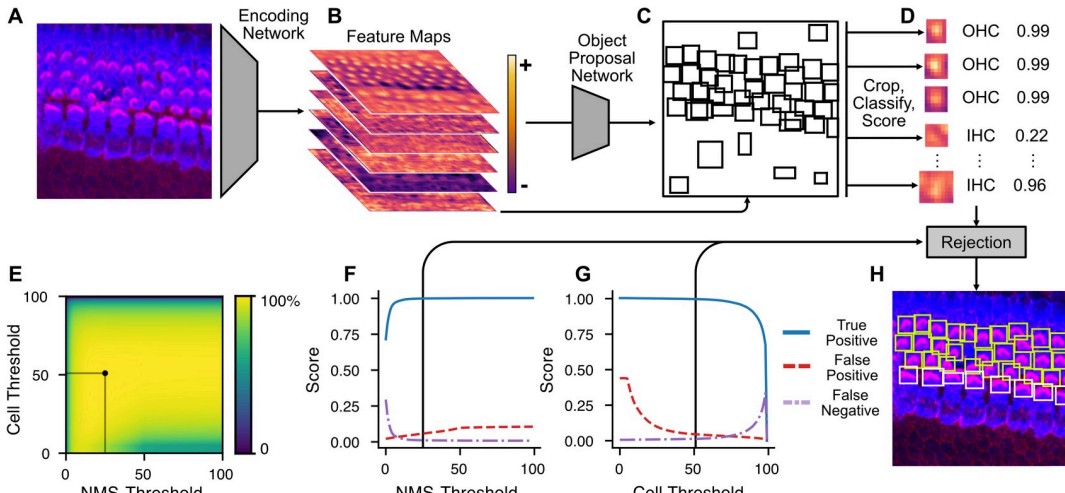

**Fig 3. Schematized overview of Faster R-CNN image detection backend.** (**A**) Input micrographs (in this case, of early postnatal mouse hair cells) are encoded into high-level representations (schematized in (**B**)) by a trained encoding convolutional neural network. These high-level representations are next passed to a region proposal network that predicts bounding boxes of objects ((**C**), schematized representation). Based on the predicted object proposals, encoded crops are classified into OHC and IHC classes by the neural network and assigned a confidence score (**D**). Next, a rejection step thresholds the resulting predictions based on confidence scores and the overlap between boxes, via NMS. Default values for user-definable thresholds were determined by the maximum average precision after a grid search of parameter combinations over eight manually annotated cochleae (**E**). The outcome of this grid search can be flattened into accuracy curves for the NMS (**F**) and rejection threshold (**G**) at their respective maxima. Boxes remaining after rejection represent the models' best estimate of each detected object in the image (**H**). IHC, inner hair cell; NMS, non-maximum suppression; OHC, outer hair cell.

process nonlinear regression [14] through the ribbon of anti-MYO7A-positive pixels, effectively treating each hair cell as a point in cartesian space. A line of best fit can be predicted through each hair cell and in doing so approximate the curvature of the cochlea. The length of this curve is then used as an approximation for the length of the cochlear coil. For example, a cell that is 20% along the length of this curve could be interpreted as one positioned at 20% along the length of the cochlea, assuming the entire cochlear coil was imaged.

To optimally perform the initial regression, individual cell detections are rasterized and then downsampled by a factor of ten using local averaging (increasing the execution speed of this step), then converted to a binary image. Next, a binary hole closing operation is used to close any gaps, and subsequent binary erosion is used to reduce the effect of nonspecific staining. Each positive binary pixel of the resulting 2D image is then treated as an X/Y coordinate that may be regressed against (**Fig 1D**). The resulting image is unlikely to form a mathematical function in cartesian space, as the cochlea may curve over itself such that for a single location on the X axis, there may be multiple clusters of cells at different Y values. To rectify this overlap, the data points are converted from cartesian to polar coordinates by shifting the points and centering the cochlear spiral around the origin, then converting each X/Y coordinate to a corresponding angle/radius coordinate. As the cochlea is not a closed loop, the resulting curve will have a gap, which is then detected by the algorithm, shifting these points by one period, and creating a continuous function. A Gaussian process [14], a generalized nonlinear function, is then fit to the polar coordinates and a line of best fit is predicted. This line is then converted back to cartesian coordinates and scaled up to correct for the earlier down-sampling (**Fig 1E**).

The apex of the cochlea is then inferred by comparing the curvature at each end of the line of best fit based on the observation that the apex has a tighter curl when mounted on a slide.

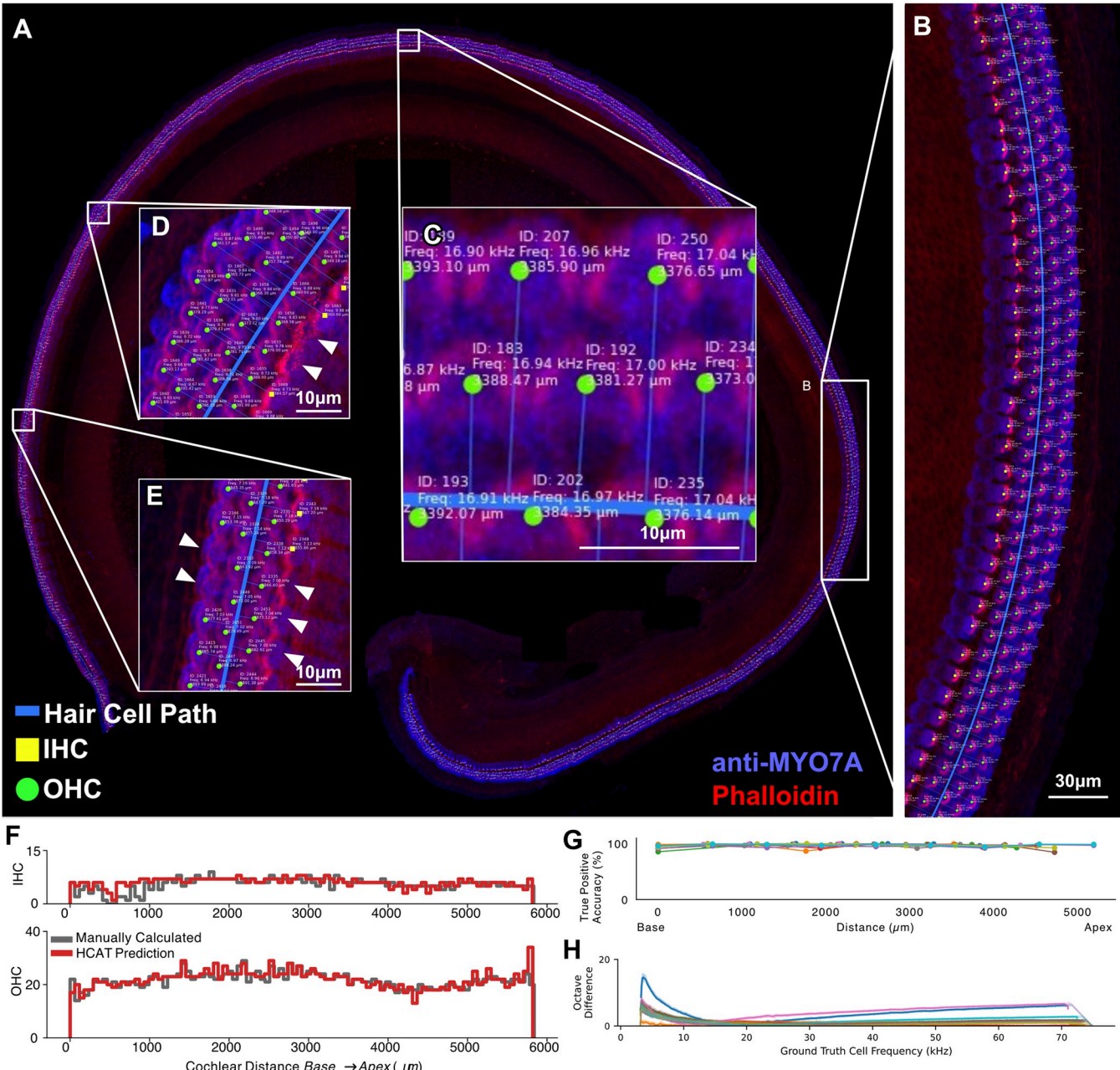

**Fig 4. Validation output of early postnatal wild-type mouse hair cell detection analysis.** A validation output image is generated for each detection analysis performed by the software. An image is automatically generated by the software similar to the one shown here for a dataset that includes an entire cochlea (**A**), with the vast majority of cells accurately detected (**B**). For each image, the model embeds information on cell's ID, its location along the cochlear coil (distance in μm from the apex), its best frequency, cell classification (IHC as yellow squares, OHC as green circles), and the line that represents tool's cochlear path estimation ((**C**), blue line). The very few examples of poor performance are highlighted in (**D**) and (**E**) (arrowheads point to three missed IHCs and two OHCs). A set of cochleograms reporting cell counts per every 1% of total cochlear length, generated with manual cell counts and frequency assignment (gray) closely agrees with an HCAT-predicted cochleogram (red) generated in a fully automated fashion (**F**). HCAT is accurate along the entire length of the cochlea (**G**), as evident by assessing the accuracy with a bin size of 10% of cochlear length. To assess the accuracy of the tool's best frequency assignment, the magnitude difference between every cell's best frequency calculated manually, and automatically, with respect to frequency for eight different cochleae is at maximum 15% of an octave across all frequencies (**H**). Each color represents one cochlea. HCAT, hair cell analysis toolbox; IHC, inner hair cell; OHC, outer hair cell.

The resulting curve closely tracks the hair cells on the image. Next, the curve's length is measured, and each detected cell is then mapped to it as a function of the total cochlear length (%). Each cell's best frequency is calculated using the Greenwood function, a species-specific method of determining cell's best frequency from its cochlear position [15] (**Fig 1F**). Upon completion of this analysis, the automated frequency assignment tool generates two cochleograms, one for IHCs and one for OHCs (**Fig 1G**).

To validate this method of best frequency assignment, we compared it to the existing standard in the field—manual frequency estimation. We manually mapped the cochlear length to cochlear frequency using a widely used ImageJ plugin, developed by the Histology Core at the Eaton-Peabody Laboratories (Mass Eye and Ear) and compared them to the results predicted by our automatic tool (**Figs 4G** and **S1**). Over 8 manually analyzed cochleae, the maximum cell frequency error of automated, relative to a manually, mapped best frequency was under 15% of an octave, with the discrepancy between the two methods less than 5% for most cells (60% of a semitone). In one cochlea, the overall cochlear path was predicted to be shorter than manually assigned, due to the threshold settings of the MYO7A fluorescence channel, causing an error at very low and very high frequencies (**Fig 4G,** dark blue). While this error was less than 15% of an octave, it is an outlier in the dataset. It is recommended, when using this tool, to evaluate the automated cochlear path estimation, and if poor, perform manual curve annotation to facilitate best frequency assignment. If required, the user is also able to switch the designation of automatically detected points representing the apical and basal ends of the cochlear coil (**Fig 1F**, red and cyan circles).

## Performance

Overall, cochleograms generated with HCAT track remarkably well to those generated manually (**Fig 4F**). Comparing HCAT to manually annotated cochlear coils (not used to train the model), we report a 98.6 ± 0.005% true positive accuracy for cell identification and a <0.01% classification error (8 cochlear coils, 4,428 IHCs and 15,754 OHCs; **S1 Fig**). We found no bias in accuracy with respect to estimated best frequency. To assess HCAT performance on a diverse set of cochlear micrographs, we sampled 88 images from 15 publications [16–30] that represent a wide variety of experimental conditions, including ototoxic treatment using aminoglycosides, genetic manipulations that could affect the hair cell anatomy, noise exposure, blast trauma, and age-related hearing loss (**Table 2**). We performed a manual quantification and automated detection analysis of these images after they were histogram-adjusted and scaled via the HCAT GUI for optimal accuracy. HCAT achieved an overall OHC detection accuracy of 98.6 ± 0.5% and an IHC detection accuracy of 96.9 ± 2.8% for 3,545 OHCs and 1,110 IHCs, with mean error of 0.34 OHC and 0.32 IHC per image. Of the 88 images we used for this validation, no errors were detected on 62 of them, and HCAT was equally accurate in images of low and high absolute cell count (**Fig 5**). Multi-piece cochleogram generation workflow is shown in (**S3 Fig**).

## Validation on published datasets

We further evaluated HCAT on whole, external datasets (generously provided by the Cunningham [31], Richardson and Kros laboratories [7]) and replicated analyses from their publications. Each dataset presented examples of organ of Corti epithelia treated with ototoxic compounds resulting in varying degrees of hair cell loss. The two datasets complement each other in several ways, covering most use cases of data analysis needs following ototoxic drug use in the organ of Corti to assess hair cell survival: in vivo versus in vitro drug application, confocal fluorescence versus widefield fluorescence microscopy imaging, early postnatal versus adult organ of Corti imaging. HCAT succeeded in quantifying the respective datasets in a fully

**Table 2. Summary of micrographs sampled from existing publications to test HCAT performance.**

| Laboratory | Number of images | OHC | IHC | Animal | Microscopy | Treatment | Age | Extent loss | Labeled protein | |
|---|---|---|---|---|---|---|---|---|---|---|
| Beurg and colleagues, 2019 [16] | 2 | 39 | 17 | Mouse | Confocal | $Tmc1^{\text{P.D569N}}$ mouse | 4 wk | — | Calbindin | Actin |
| Fang and colleagues, 2019 [17] | 1 | 42 | 14 | Mouse | Confocal | WT mouse | 6–8 wk | — | MYO7A | Actin |
| Fu and colleagues, 2021 [18] | 6 | 175 | 69 | Mouse | Confocal | $Klc2^{-/-}$ mouse | P50 | — | MYO7A | Actin |
| Gyorgy and colleagues, 2019 [19] | 11 | 330 | 113 | Mouse | Confocal | $Tmc1^{Bth}$ mutant | 24 wk | — | MYO7A | Actin |
| He and colleagues, 2021 [20] | 7 | 304 | 69 | Mouse | Confocal | Noise trauma | 14 wk | 0% | Calcineurin; 4-HNE | Actin |
| Hill and colleagues, 2016 [21] | 5 | 171 | 0 | Mouse | Confocal | Noise trauma | 16 wk | 0% | p-AMPKα | Actin |
| Kim and colleagues, 2018 [22] | 3 | 102 | 33 | Mouse | Confocal | Blast trauma | 6 wk | 0–55% | MYO7A | Actin |
| Lee and colleagues, 2017 [23] | 2 | 24 | 7 | Mouse | Confocal | WT mouse | E12-E17 | — | MYO7A | Actin |
| Li and colleagues, 2020 [24] | 2 | 70 | 21 | Mouse | Confocal | $Myo7a\text{-}\Delta C$ mouse | 9 wk | — | MYO7A | Actin |
| Mao and colleagues, 2021 [25] | 9 | 916 | 311 | Mouse | Confocal | Blast trauma | 10 wk | 0–69% | MYO7A | Actin |
| Sang and colleagues, 2015 [26] | 6 | 193 | 65 | Mouse | Confocal | $Idlr1^{-/-}$ mouse | P28 | — | MYO7A | Actin |
| Sethna and colleagues, 2021 [27] | 7 | 274 | 104 | Mouse | Confocal | $Pcdh15^{R250X}$ mouse | P60 | — | MYO7A | Actin |
| Wang and colleagues, 2011 [28] | 2 | 90 | 26 | Mouse | Confocal | $SCX^{-/-}$ mouse | P18 | — | MYO7A | Actin |
| Yousaf and colleagues, 2015 [29] | 24 | 760 | 244 | Mouse | Confocal | $Map3k1^{tm1Yxia}$ | P90 | — | MYO7A | Actin |
| Zhao and colleagues, 2021 [30] | 1 | 55 | 17 | Mouse | Confocal | $Clu^{-/-}$ mouse | 9mo | — | MYO7A | Actin |
| Total | 88 | 3,545 | 1,110 | | | | | | | |

HCAT, hair cell analysis toolbox; IHC, inner hair cell; OHC, outer hair cell.

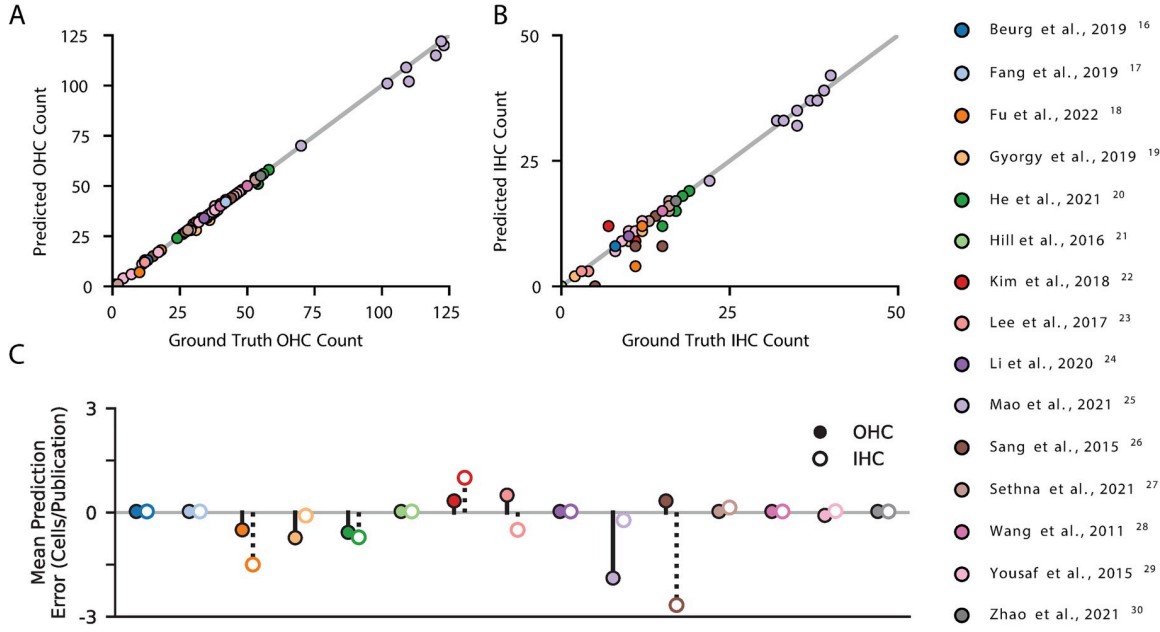

**Fig 5. HCAT detection performance on published images of cochlear hair cells.** HCAT detection performance was assessed by running a cell detection analysis in the GUI on 88 confocal images of cochlear hair cells sampled from published figures across 15 different original studies [16–30]. None of the images from this analysis were used to train the model. Each image was adjusted within the GUI for optimal detection performance. Cells in each image were also manually counted (presented as ground truth values) and results compared to HCAT's automated detection. The resulting population distributions of hair cells are compared for OHCs (**A**), and IHCs (**B**). The mean difference in predicted number of IHCs (open circles) and OHCs (filled circles) in each publication is summarized for each cell type: zero indicates an accurate detection, negative values indicate false negative detections, while positive values indicate false positive detections (**C**). GUI, graphical user interface; HCAT, hair cell analysis toolbox; IHC, inner hair cell; OHC, outer hair cell.

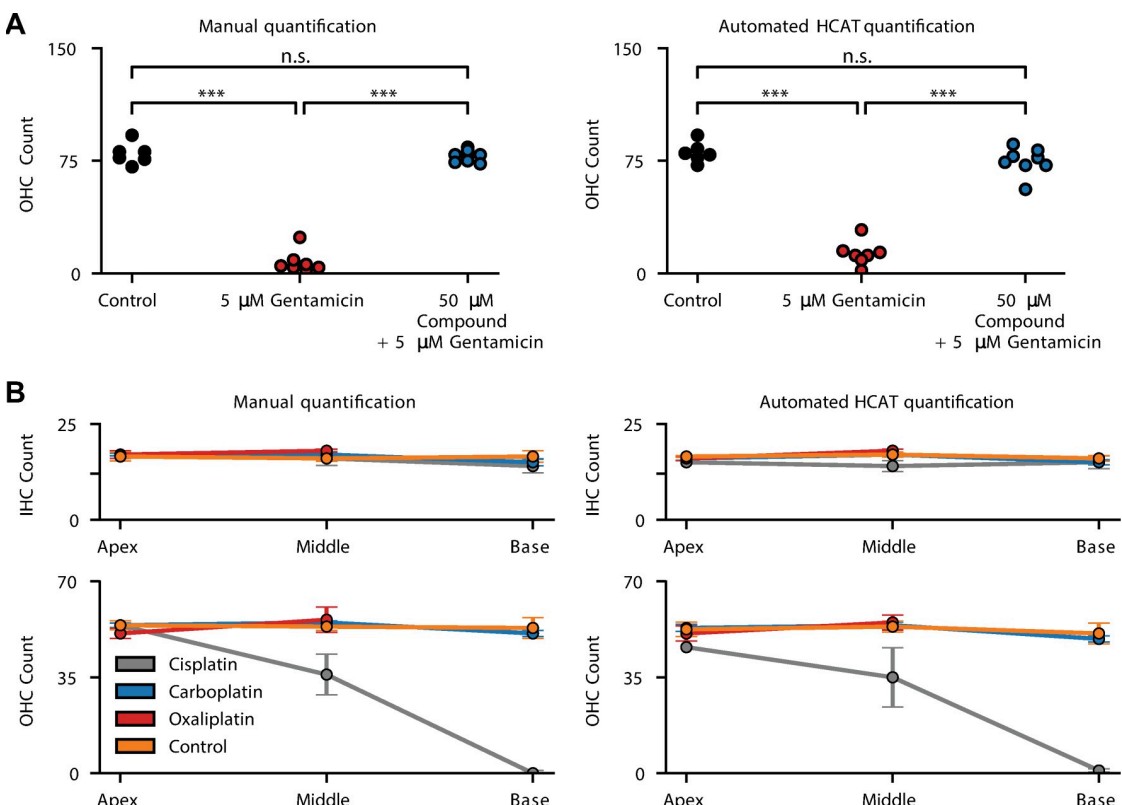

**Fig 6. Evaluation of HCAT performance on cochlear datasets to assess ototoxic drug effect.** To assess HCAT performance on aberrated cochlear samples, we compared HCAT analysis results to manual quantification on datasets from two different publications focused on assessing hair cell survival following treatment with ototoxic compounds. **(A)** Original imaging data of P3 CD1 mouse cochlea, underlying the finding in Fig 2F of Kenyon and colleagues, 2021 [7], generously provided by the Richardson and Kros laboratories. Images were collected using epifluorescence microscopy, following a 48-h incubation in either 0 μm gentamicin (Control), 5 μm gentamicin, or 5 μm gentamicin + 50 μm test compound UoS-7692. Each symbol represents the number of OHCs in a mid-basal region from 1 early postnatal in vitro cultured cochlea [7]. One-way ANOVA with Tukey's multiple comparison tests. ***, $p < 0.001$; $ns$, not significant. In some cases, HCAT detections overestimated the total number of surviving hair cells in the gentamycin-treated tissue. However, overall, the software-generated results are in agreement with those of the original study, drawing the same conclusion. **(B)** Original imaging data of adult mouse cochleae, underlying the finding in Fig 7A-B in Gersten and colleagues, 2020 [31] were generously provided by the Cunningham laboratory. In this study, mice were treated by in vivo application of clinically proportional levels of ototoxic compounds, Cisplatin, Carboplatin, Oxaliplatin, and Saline (control), in an intraperitoneally cyclic delivery protocol [31]. Regions of interest were imaged at the base, middle, and apex of each cochlea. HCAT's automated detections were comparable to manual quantification and were sufficient to draw a conclusion that is consistent with the original publication. Upon comparison, HCAT had higher detection accuracy in OHCs, compared to IHCs, likely due to the variability of the MYO7A fluorescence intensity levels in IHCs across the dataset. HCAT, hair cell analysis toolbox; IHC, inner hair cell; OHC, outer hair cell.

automated fashion with an accuracy sufficient to replicate the main finding in each study (**Fig 6**), underestimating the total number of cells for Gersten and colleagues, 2020 by 7.3%, and overestimating the total number of cells from Kenyon and colleagues, 2021 by 1.47%. It is worth noting that these datasets were collected without optimization for an automated analysis. Thus, we expect an even higher performance accuracy with an experimental design optimized for HCAT-based automated analysis.

## Discussion

Here, we present the first fully automated cochlear hair cell analysis pipeline for analyzing multiple micrographs of cochleae, quickly detecting and classifying hair cells. HCAT can

analyze whole cochleae or individual regions and can be easily integrated into existing experimental workflows. While there were previous attempts at automating this analysis, each were limited in their use to achieve widespread application [3,5]. HCAT allows for unbiased, automated hair cell analysis with detection accuracy levels approaching that of human experts at a speed so significantly faster that it is desirable even with rare errors. Furthermore, we validate HCAT on data from various laboratories and find it is accurate across different imaging modalities, staining, age, and species.

Deep learning-based detection infers information from the pixels of an image to make decisions about what objects are and where they are located. To this end, the information is devoid of any context. HCAT's deep learning detection model was trained largely using anti-MYO7A and phalloidin labels; however, the model can perform on specimens labeled with other markers, as long as they are visually similar to examples in our training data. For example, some of the validation images of cochlear hair cells sampled from published figures contained cell body label other than MYO7A, such as Calbindin [16,32], Calcineurin [20,33], and p-AMPKα [34], while in other images, phalloidin staining of stereocilia bundle was substituted by anti-espin [35] labeling. Although no images containing hair cell-specific nuclear markers, such as pou4f3 [36], were included in the pool training data, HCAT performed reasonably well when tested on such images, especially when they also contained a bundle stain. Of higher importance is the quality of the imaging data: proper focus adjustment, high signal-to-noise ratio, image resolution, and adequately adjusted brightness and contrast settings. Furthermore, the quality of the training dataset greatly affects model performance; upon validation, HCAT performed slightly worse when evaluated on community provided datasets due to fewer representative examples within the pool of our training data.

We will strive to periodically update our published model when new data arise, further improving performance over time. At present, HCAT has proven to be sufficiently accurate to consistently replicate major findings even with occasional discrepancies to a manual analysis, even when used on datasets that were collected without any optimization for automated analysis. The strength of this software is in automation, allowing for processing thousands of hair cells over the entire cochlear coil without human input. Recent advancements in tissue-clearing techniques enable the acquisition of the intact 3D architecture of the cochlear coil using confocal or two-photon laser scanning microscopy allowing for future development of the HCAT tool as the wealth of such imaging data are made available to the public. Although no tissue-cleared data were used to develop HCAT, we tested it on few published examples of tissue-cleared mouse and pig cochlear imaging data [3,4]. While HCAT showed reasonable hair cell detection rates, the tool was unable to perform as accurate as we report for high-resolution confocal imaging data, most likely because the tissue-cleared datasets were collected at lower resolution (0.65 to 0.99 μm/pix), and contained only anti-MYO7a fluorescence.

It is common for the population of missing cells, rather than absolute counts, to be reported in cell survival studies. We were unable to support missing cell detection or quantification in HCAT. We found there lacked sufficient, and robust information on the locations of missing cells to automate their detection consistently and accurately. In some cases, a distinctive "X-shaped" phalangeal scar may be seen in the sensory epithelium following hair cell loss [37,38] that may be sufficient to determine the presence of a missing cell; however, this is often visible with an actin stain or on scanning electron microscopy images, and not so in the other pathologic cases HCAT attempts to support.

While the detection model was trained and cochlear path estimation designed specifically for cochlear tissue, HCAT can serve as a template for deep learning-based detection tasks in other types of biological tissue in the future. While developing HCAT, we employed best practices in model training, data annotation, and augmentation. With minimal adjustment and a

small amount of training data, one could adapt the core codebase of HCAT to train and apply a custom deep learning detection model for any object in an image.

To our knowledge, this is the first whole cochlear analysis pipeline capable of accurately and quickly detecting and classifying cochlear hair cells. HCAT enables expedited cochlear imaging data analysis while maintaining high accuracy. This highly accurate and unsupervised data analysis approach will both facilitate ease of research and improve experimental rigor in the field.

## Materials and methods

### Preparation and imaging of in-house training data

Organs of Corti were dissected in one contiguous piece at P5 in Leibovitz's L-15 culture medium (21083–027, Thermo Fisher Scientific) and fixed in 4% formaldehyde for 1 h. The samples were permeabilized with 0.2% Triton-X for 30 min and blocked with 10% goat serum in calcium-free HBSS for 2 h. To visualize the hair cells, samples were labeled with an anti-Myosin 7A antibody (#25–6790 Proteus Biosciences, 1:400) and goat anti-rabbit CF568 (Biotium) secondary antibody. Additionally, samples were labeled with Phalloidin to visualize actin filaments (Biotium CF640R Phalloidin). Samples were then flattened into one turn, mounted on slides using ProLong Diamond Antifade Mounting kit (P36965, Thermo Fisher Scientific), and imaged with a Leica SP8 confocal microscope (Leica Microsystems) using a 63×, 1.3 NA objective. Confocal Z-stacks of 512 × 512 pixel images with an effective pixel size of 288 nm were collected using the tiling functionality of the Leica LASX acquisition software and maximum intensity projected to form 2D images. All experiments were carried out in compliance with ethical regulations and approved by the Animal Care Committee of Massachusetts Eye and Ear.

### Training data

Varied data are required for the training of generalizable deep learning models. In addition to imaging data collected in our lab, we sourced generous contributions from the larger hearing research community from previously reported [7,31,39–46], and in some cases unpublished, studies. Bounding boxes for hair cells seen in maximum intensity projected z-stacks were manually annotated using the labelImg [47] software and saved as an XML file. For whole cochlear cell annotation, a "human in the loop" approach was taken, first evaluating the deep learning model on the entire cochlea, visually inspecting it, then manually correcting errors. Our dataset contained examples from three different species, multiple ages, microscopy types, and experimental conditions. Only the images generated in-house contain an entire, intact cochlea. A summary of our training data is presented in **Table 1**.

### Training procedure

The deep learning architectures were trained with the AdamW [48] optimizer with a learning rate starting at $1 \times 10^{-4}$ and decaying based on cosine annealing with warm restarts with a period of 10,000 epochs. In cases with a small number of training images, deep learning models tend to fail to generalize and instead "memorize" the training data. To avoid this, we made heavy use of image transformations that randomly add variability to the original set of training images and synthetically increase the variety of our training datasets [49] (**S2 Fig**).

### Hyperparameter optimization

Eight manually annotated cochleae were evaluated with the Faster R-CNN detection algorithm without either rejection method (via detection confidence or non-maximum suppression). A

grid search was performed by breaking each threshold value into 100 steps from zero to one, and each combination applied to the resulting cell detections, reducing their number, then calculating the true positive (TP), true negative (TN), and false positive (FP) rates (**S1D and S1E Fig**). An accuracy metric of the TP minus both TN and FP was calculated and averaged for each cochlea. The combinations of values that produce the highest accuracy metric were then chosen as default for the HCAT algorithm.

## Computational environment

HCAT is operating system agnostic, requires at least 8 GB of system memory, and optionally an NVIDIA GPU with at least 8 GB of video memory to optional GPU acceleration. All scripts were run on an analysis computer running Ubuntu 20.04.1 LTS, an open-source Linux distribution from Canonical based on Debian. The workstation was equipped with two Nvidia A6000 graphics cards for a total of 96 GB of video memory. Many scripts were custom written in python 3.9 using open-source scientific computation libraries including numpy [50], matplotlib, and scikit-learn [51]. All deep learning architectures, training logic, and much of the data transformation pipeline was written in pytorch [52] and making heavy use of the torchvision [52] library.

## Supporting information

**S1 Fig. Validation of hair cell detection analysis and location estimation.** Whole cochlear turns (**A**) were manually annotated and evaluated with the HCAT detection analysis pipeline. Each analysis generated cochleograms (**B**), reporting the "ground truth" result obtained from manual segmentation (dark lines) superimposed onto the cochleogram generated from hair cells detected by the HCAT analysis (light lines). The best frequency estimation error was calculated as an octave difference of predicted best frequency for every hair cell versus their manually assigned frequency using the ImageJ plugin (**C**). Optimal cell detection and non-maximum suppression thresholds were discerned via a grid search by maximizing the true positive rate penalized by the false positive and false negative rates (**D**). Black lines on the curves (**E**) denote the optimal hyperparameter value.
(EPS)

**S2 Fig. Training data augmentation pipeline.** Training images underwent data augmentation steps, increasing the variability of our dataset and improving resulting model performance. Examples of each transformation are shown on exemplar grids (bottom). Each of these augmentation steps was probabilistically applied sequentially (left to right, as shown by arrows) during every epoch.
(EPS)

**S3 Fig. Multi-piece cochleogram generation workflow for HCAT.** Adult murine cochlear dissection, depending on technique used, typically produces up to 6 individual pieces of tissue, numbered from apex to base in (**A**). These pieces form the entirety of the organ of Corti and can be analyzed by HCAT (**B**). First, each piece must have its curvature annotated manually in ImageJ from base to apex (**C**) using the EPL cochlea frequency ImageJ plugin. Then, these annotations and images are passed one-at-a-time to the HCAT command line interface. This will generate a CSV for each file, which are then manually compiled (**E**). This allows for the generation of a complete cochleogram from a multi-piece dissection (**F**).
(EPS)

**S1 Data.** A compressed folder with spreadsheets containing, in separate files, the underlying numerical data and statistical analysis for Figs 1G, 3E, 3F, 3G, 4F, 4H, 5A, 5B, 5C, 6A, 6B, S1B,

S1C, S1D, S1E, S1F.
(ZIP)

## Acknowledgments

We would like to thank Dr. Marcelo Cicconet (Image and Data Analysis Core at Harvard Medical School) and Haobing Wang, MS (Mass Eye and Ear Light Microscopy Imaging Core Facility) for their assistance in this project. We thank Dr. Lisa Cunningham, Dr. Michael Deans, Dr. Albert Edge, Dr. Katharine Fernandez, Dr. Ksenia Gnedeva, Dr. Yushi Hayashi, Dr. Tejbeer Kaur, Dr. Jinkyung Kim, Prof. Corne Kros, Dr. M. Charles Liberman, Dr. Vijayprakash Manickam, Dr. Anthony Ricci, Prof. Guy Richardson, Dr. Mark Rutherford, Dr. Basile Tarchini, Dr. Amandine Jarysta, Dr. Bradley Walters, Dr. Adele Moatti, Dr. Alon Greenbaum, the members of their teams and all other research groups, for providing their datasets to evaluate the HCAT. We thank Hidetomi Nitta, Emily Nguyen, and Ella Wesson for their assistance in generating a portion of training data annotations. We also thank Evan Hale and Corena Loeb for critical reading of the manuscript.

## Author Contributions

**Conceptualization:** Christopher J. Buswinka, Artur A. Indzhykulian.

**Data curation:** Christopher J. Buswinka, Richard T. Osgood, Rubina G. Simikyan.

**Formal analysis:** Christopher J. Buswinka, Richard T. Osgood, Artur A. Indzhykulian.

**Funding acquisition:** Artur A. Indzhykulian.

**Investigation:** Christopher J. Buswinka, David B. Rosenberg, Artur A. Indzhykulian.

**Methodology:** Christopher J. Buswinka, Artur A. Indzhykulian.

**Project administration:** Artur A. Indzhykulian.

**Resources:** Christopher J. Buswinka, Artur A. Indzhykulian.

**Software:** Christopher J. Buswinka.

**Supervision:** Christopher J. Buswinka, Artur A. Indzhykulian.

**Validation:** Christopher J. Buswinka, Richard T. Osgood, Artur A. Indzhykulian.

**Visualization:** Christopher J. Buswinka, Richard T. Osgood, Artur A. Indzhykulian.

**Writing – original draft:** Christopher J. Buswinka, Artur A. Indzhykulian.

**Writing – review & editing:** Richard T. Osgood, Rubina G. Simikyan, David B. Rosenberg.

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
