## [Editor Report · Decision Letter 0]

18 Nov 2022

Dear Dr Indzhykulian, 

Thank you for submitting your manuscript entitled "The Hair Cell Analysis Toolbox: A machine learning-based whole cochlea analysis pipeline" for consideration as a Methods and Resources by PLOS Biology.

Your manuscript has now been evaluated by the PLOS Biology editorial staff, as well as by an academic editor with relevant expertise, and I am writing to let you know that we would like to send your submission out for external peer review.

However, before we can send your manuscript to reviewers, we need you to complete your submission by providing the metadata that is required for full assessment. To this end, please login to Editorial Manager where you will find the paper in the 'Submissions Needing Revisions' folder on your homepage. Please click 'Revise Submission' from the Action Links and complete all additional questions in the submission questionnaire. Please note that we do consider and appreciate suggested reviewers and their expertise, and we permit exclusions of up to 3 individuals.

Once your full submission is complete, your paper will undergo a series of checks in preparation for peer review. After your manuscript has passed the checks it will be sent out for review. To provide the metadata for your submission, please Login to Editorial Manager (https://www.editorialmanager.com/pbiology) within two working days, i.e. by Nov 20 2022 11:59PM.

Kind regards,

Kris

Kris Dickson, Ph.D., (she/her)

Neurosciences Senior Editor/Section Manager

PLOS Biology

kdickson@plos.org

---

## [Decision Letter · Decision Letter 1]

9 Jan 2023

Dear Dr Indzhykulian,

Thank you for your patience while your manuscript "The Hair Cell Analysis Toolbox: A machine learning-based whole cochlea analysis pipeline" went through peer-review at PLOS Biology as a Methods and Resources submission. Your manuscript has now been evaluated by the PLOS Biology editors, an Academic Editor with relevant expertise, and by several independent reviewers.

In light of the reviews, which you will find at the end of this email, we are pleased to offer you the opportunity to address the comments from the reviewers in a revision that we anticipate should not take you very long. We will then assess your revised manuscript and your response to the reviewers' comments with our Academic Editor aiming to avoid further rounds of peer-review, although might need to consult with the reviewers, depending on the nature of the revisions.

**IMPORTANT - SUBMITTING YOUR REVISION**

*Resubmission Checklist*

Please make sure to read and address the following important policies and guidelines while preparing your revision. Failure to address all of these points will delay further handling of your submission:

*Published Peer Review*

*PLOS Data Policy*

Please note that as a condition of publication PLOS' data policy (http://journals.plos.org/plosbiology/s/data-availability) requires that you make available all data used to draw the conclusions arrived at in your manuscript. If you have not already done so, you must include any data used in your manuscript either in appropriate repositories (e.g. Zenodo, Figshare, etc), within the body of the manuscript, or as supporting information (N.B. We DO NOT require all raw data, rather was request all of the summary data used to generate the figures be included as these numerical values that were used to generate graphs, histograms etc are necessary for others to reproduce your work). 

For an example see here: http://www.plosbiology.org/article/info%3Adoi%2F10.1371%2Fjournal.pbio.1001908#s5

Exceptions to this policy are also in place for third party data sharing, but please explicitly state where this is the case.

Please provide summary data for the following graphs:

Fig1G; Fig3C,F,G, Fig4F,G; *Fig5A-C (note that we only need summary data, not raw data), Fig6A,B;

Supplemental Fig1B-E;

Sincerely,

Kris

Kris Dickson, Ph.D., (she/her)

Neurosciences Senior Editor/Section Manager

PLOS Biology

kdickson@plos.org

***EDITORIAL REQUESTS:

1) Consider a title change to more clearly convey the utility of this tool:

The Hair Cell Analysis Toolbox is a precise and fully-automated pipeline for analyses of whole cochlea across species

OR

The Hair Cell Analysis Toolbox is a fast, precise and fully-automated machine learning-based pipeline for analyses of whole cochlea across species

OR

The Hair Cell Analysis Toolbox: A fast, precise and fully-automated machine-learning-based pipeline for analyses of whole cochlea across species

2) Consider a somewhat pared down abstract, removing some of the background and focusing more on the interesting findings in this work:

Our sense of hearing is mediated by sensory hair cells, precisely arranged and highly specialized cells subdivided into outer hair cells (OHCs) and inner hair cells (IHCs). Microscopy tools allow for imaging of auditory hair cells along the full length of the cochlea, often yielding more data than feasible to manually analyze. Currently, there are no widely applicable tools for fast, unsupervised, unbiased, and comprehensive image analysis of auditory hair cells that work well either with imaging datasets containing an entire cochlea or smaller sampled regions. Here, we present a highly-accurate machine learning-based hair cell analysis toolbox for the comprehensive analysis of whole cochleae (or smaller regions of interest) across imaging modalities and species. The Hair Cell Analysis Toolbox (HCAT) is a software that automates common image analysis tasks such as counting hair cells, classifying them by subtype (IHCs vs OHCs), determining their best frequency based on their location along the cochlea, and generating cochleograms. These automated tools remove a considerable barrier in cochlear image analysis, allowing for faster, unbiased, and more comprehensive data analysis practices. Furthermore, HCAT can serve as a template for deep-learning based detection tasks in other types of biological tissue: with some training data, HCAT's core codebase can be trained to develop a custom deep learning detection model for any object on an image.

3) Please go through your submission and make sure to clearly state the species being referred to in each instance within the results and in the methods.

***REVIEWS:

Reviewer's Responses to Questions

Do you want your identity to be public for this peer review?

Reviewer #1: No

Reviewer #2: Yes: Shigeo Okabe

Reviewer #3: Yes: Alan Cheng

Reviewer #1: The Hair Cell Analysis Toolbox: A machine learning-based whole cochlea 1 analysis pipeline.

2 Christopher J. Buswinka, Richard T. Osgood, Rubina G. Simikyan, David B. Rosenberg, Artur A. Indzhykulian 3 Mass Eye and Ear, Harvard Medical School.

In this paper the authors present a method, 'HCAT', for automated analysis of cochlear tissue, demonstrating accurate replication of 'ground truth'; manually analyzed counts of outer and inner hair cells. HCAT reduces the worktime by orders of magnitude, eliminates observer bias and subjectivity, and provides objective criteria for comparison between studies. Quantified cochlear structure is needed to study function, dysfunction, and therapeutic advances. This is the case for preclinical animal studies, and potentially for application to human temporal bone collections where health history can then be correlated with hair cell loss. Impressively, the authors not only validate HCAT against their own, and published cochlear images, but do so as well for tissue subjected to ototoxic damage. In all cases HCAT performs extremely well. This analytical tool will be of great utility to the hearing research community, and speed the advance of both experimental and clinical (cadaveric) studies. 

The paper is well-written. A few points were noted about figures and tables and those follow. 

Figure 1. Although it's more or less obvious that the paper is about the mouse cochlea, this should be stated in the Methods, including genotype, and tissue source should be identified in each Figure legend including species and age. 

Figure 1. Scale bars for panels A, B, C? Better still, mark off distance in microns along the entire length of image in Panel A. 

How are multiple turns flattened to obtain panel A? The mouse cochlea has approximately 2.5 turns in situ. How does the tissue become a single arc? Is this by shaping after dissection? Or is this from the image analysis? If from the analysis, is it possible to illustrate diagrammatically? Not absolutely necessary, but would help the Reader understand the process. 

Panel A lists phalloidin to label stereocilia as blue, and anti-myo7A to label hair cell somata as red. While it's not possible to distinguish in panel A, in panel B it appears that the stereociliary bundles are red (ish) and the hair cells bodies are blue (ish). Labels reversed? 

Figure 2A. IHCs and OHCs pseudo-colored? Or are these the little squares at low mag?? 

Table 1 - # of images constituting what total cochlear length? This would help to evaluate whether each observer found the same hair cell density (i.e., per 100 microns). 

Figure 3. "encoded crops are classified into OHC and IHC classes" manually?

Lines 168-169 versus lines 174-175. Should these values be the same? 

Line 177 - 0.15%, typo?

Table 2 - total cochlear length in images? 

Have the authors considered analysis for 'subcellular' biomarkers such as pre- and postsynaptic proteins? Some immunolabels have very high signal to noise (e.g., anti-CtBP2, anti-synapsin, anti-Na\\K ATPase). 

Reviewer #2: This manuscript reports a new computational toolbox for the comprehensive analysis of whole cochleae or their smaller regions for counting the number of both inner and outer hair cells, their location, and their best frequency. There are several previous computational tools for automatic analysis of hair cell analysis in the cochlea, but the reported toolbox has advantages in efficiently detecting sensory cells using deep-learning-based detection techniques. This algorithm will help researchers in the field of hearing research by providing a computational tool applicable to samples with variable dissection, fixation, and staining conditions. However, several points should be corrected and improved in a future manuscript.

Major points.

1. This toolbox can accept two-dimensional data (projection images) of immunostained cochleae. Recent advancements in tissue-clearing techniques enabled the acquisition of the intact 3D structure of the cochlea using confocal or two-photon laser scanning microscopy. Such 3D image data require programs different from the toolbox presented in this manuscript and will become more critical in the realistic modeling of sound propagation through the cochlear duct. This point should be clarified in the introduction part.

2. The toolbox can reliably detect the position of outer and inner hair cells (OHC and IHC)in the middle part of the organ of Corti. However, it may be difficult to identify OHC and IHC at both ends of the structure (apical and basal ends of the organ of Corti). Therefore, information about the reliability of cell detection at the two ends should be presented.

3. The samples analyzed in this study mainly came from manually dissected organs of Corti at P5. The dissection of the adult organ of Corti is challenging and, in many cases, results in fragmentation of the structure. Can this protocol be applied to the analysis of the whole cochlear structure from the adult mouse? If this is feasible, please show the examples.

Minor points.

1. For the analysis of cochlear pathology, the positions of lost OHCs/IHCs are more important than the remaining OHCs/IHCs. The authors presented the loss of a small number of cells in Figure 4E. However, it is unclear if the toolbox can identify the original locations of lost cells in the cochlea after noise-induced hearing loss, which often shows massive cell loss at the specific locations along the longitudinal axis of the organ of Corti. It is desirable to show the toolbox's ability to find the position of lost cells.

2. In Table 1, please show the age of the animals.

3. In Table 2, please show the extent of cell loss for samples with noise trauma. 

4. In Supplemental Figure 1, the sample No. 4 with the straightened sensory epithelium showed the highest frequency error. The relationship between the sample deformation and the frequency error should be discussed. Is this sample No.4 different from the one mentioned in the text ("due to the threshold settings of the MYO7A channel, causing an error at very low and very high frequencies")?

Reviewer #3: The submitted manuscript describes a unique and throughput method of quantifying hair cells which will be useful in many studies. Quantifying hair cells accurately in an automated manner has been challenging in part because of the proximity of adjacent cochlear hair cells. So this can be a valuable addition to our field. By using immunostaining for myo7a and hair bundles (phalloidin), and subsequently against other markers (eg. Calb and esp), the authors describe the successful application of this method to a variety of published studies and data from other labs. Overall the paper is straightforward and easy to follow. While the overall conclusion seems supported, the paper will benefit from clarification of the methods, potential limitation and showing examples of validation. I only several minor comments to help improve the paper.

-They mention in the paper that other markers can be used like Calb and Esp so it would be nice to see a representative example of what that and the ROI would look like using those (or other) markers. 

-can the method be used to count hair cells without bundles?

-how do they distinguish between inner and outer hair cells. Is it based on phalloidin pattern or localization? 

-L126 they look at ototoxic induced cochleae from different labs which I think was a good compliment to the study. However, there's no representative images of these and how they were quantified. It would be nice to see how these ROIs were done with damage visually even though they do quantify. 

-can the method be applied to a nuclear hair cell stain? E.g. pou4f3?

-if the location of hair cells is determined by curvature of the cochlear turn, doesn't that depend on how tissues are mounted? If so, that seems problematic.

-L214-It would have been nice to have a % accuracy here too, rather than the vaguely worded "sufficient to replicate main finding"

-L276-This seems unnecessary and poorly worded, please remove

---

## [Editor Report · Decision Letter 2]

17 Feb 2023

Dear Dr Indzhykulian,

Thank you for the submission of your revised Methods and Resources "The Hair Cell Analysis Toolbox is a precise and fully-automated pipeline for whole cochlea hair-cell quantification." for publication in PLOS Biology. On behalf of my colleagues and the Academic Editor, Andy Groves, I am pleased to say that we can in principle accept your manuscript for publication, provided you address any remaining formatting and reporting issues. These will be detailed in an email you should receive within 2-3 business days from our colleagues in the journal operations team; no action is required from you until then. Please note that we will not be able to formally accept your manuscript and schedule it for publication until you have completed any requested changes.

PRESS

Sincerely, 

Kris Dickson, Ph.D., (she/her)

Neurosciences Senior Editor/Section Manager

PLOS Biology

kdickson@plos.org